# Epidemiological factors associated with immunological resistance in household contacts exposed to active tuberculosis in South Africa: A logistic regression analysis

Nomfanelo Maenetje[1,2]*, Matthew Oladimeji[1], Mandla Mlotshwa[1,3,4], Daniel Hoft[5], Christina Lindan[6], Robert Wallis[1,3], Salome Charalambous[1,3,7], Gavin Churchyard[1,7], Vinodh Edward[1,3], Andrew Fiore-Gartland[8], Jerry Shai[2], Pholo Maenetje[1,3]

1 The Aurum Institute, Johannesburg, South Africa, 2 Department of Biomedical Sciences, Tshwane University of Technology, Pretoria, South Africa, 3 Department of Medicine, Vanderbilt University, Nashville, United States of America, 4 Faculty of Health Sciences, Department of Biomedical Sciences, University of Johannesburg, Johannesburg, South Africa, 5 Departments of Internal Medicine and Molecular Microbiology and Immunology, Saint Louis University, St. Louis, Missouri, United States of America, 6 Department of Epidemiology and Biostatistics, University of California at San Francisco, San Francisco, California, United States of America, 7 School of Public Health, Faculty of Health Sciences, University of the Witwatersrand, Johannesburg, South Africa, 8 HIV Vaccine Trials Network, Fred Hutchinson Cancer Research Center, Seattle, United States of America

* nmaenetje@auruminstitute.org

## Abstract

### Introduction

Studying individuals who do not get infected with tuberculosis (TB) despite being persistently exposed to infectious TB may enable us to identify TB protective mechanisms.

### Methods

Between Apr 2015 and Apr 2017, we recruited adult household contacts (HHCs) of index TB cases (GeneXpert and sputum smear-positive) in Rustenburg, South Africa. HIV-uninfected HHCs who tested positive on both Tuberculin Skin Test (TST) and QuantiFERON-TB Gold In-tube (QFT) were defined as having latent TB infection (QFT+TST+), and those who tested double negative (QFT-TST-) were defined as uninfected with TB. The level of risk for TB infection was evaluated using an epidemiologic risk score. We compared epidemiological and clinical characteristics between the groups and used logic regression to identify factors associated with being QFT-TST-.

### Results

Of the 235 household contacts screened, 109 (46.3%) were QFT+TST+, 46 (19.5%) were TST-QFT-, 73 (30.1%) had discordant results, and 7 (2.9%) were excluded

**Data availability statement:** All relevant data are within the paper and its Supporting Information files.

**Funding:** The research reported in this publication was made possible through funding by the Strategic Health Innovation Partnerships (SHIP) Unit of the South African Medical Research Council. This research was also supported by the Sub-Saharan African Network for TB/HIV Research Excellence (SANTHE) which is funded by the Science for Africa Foundation [Del-22-007] with support from Wellcome Trust and the UK Foreign, Commonwealth & Development Office and is part of the EDCPT2programme supported by the European Union; the Bill & Melinda Gates Foundation [INV-033558]; and Gilead Sciences Inc., [19275]. All content contained within is that of the authors and does not necessarily reflect positions or policies of any SANTHE funder. For the purpose of open access, the author has applied a CC BY public copyright license to any Author Accepted Manuscript version arising from this submission. Training in scientific manuscript preparation and publication was provided to Nomfanelo Maenetje through the International Traineeships in AIDS Prevention Studies (ITAPS) Program, of the University of California, San Francisco's (funded by U.S. NIMH, R25MH064712 [2014–2019], R25123256 [2020–2025] and IAVI).

**Competing interests:** The authors declared that no competing interests exist.

based on being HIV positive, already having active TB disease or had missing QFT/TST results. After 3 months, 27 (58.6%) of HHCs remained persistently negative. Younger age, higher number of household windows and habitable rooms, and relations with the index case were independently associated with being QFT-TST-. In the multivariable analysis, younger age (OR: 2.81, 95% CI, 1.23–6.47) and living in homes with more rooms (OR: 4.62, 95% CI, 1.81–11.79) remained associated with being QFT-TST-. We found no association between QFT-TST- and factors such as time spent with the index case, type of house, number of household occupants, or the risk score.

## Conclusion

Our findings that both younger age and larger living quarters were associated with QFT-TST- status may suggest reduced exposure to TB. We found no association between the epidemiological TB risk score consisting of multiple TB infection risk factors and QFT-TST- status, suggesting other factors may play a role in remaining TB uninfected despite exposure.

---

## Introduction

In 2022, an estimated 10.6 million new cases of TB (range, 9.9–11.4 million) and 1.3 million TB-associated deaths occurred globally [1], making TB one of the leading causes of death by an infectious agent. South Africa is among the top countries that account for two-thirds of the global TB epidemic, with approximately 280 000 new TB cases reported in 2022, resulting in nearly 54 000 deaths from TB disease [1]. While approximately 23–25% of individuals are infected with latent TB globally [2], latent TB rates in South Africa are much higher, with an estimated 80% of adults having latent TB [3]. In the past few years, there has been an interest in studying a small number of individuals who remain TB uninfected or do not acquire latent TB despite high and sustained exposure to TB [4–6]. These individuals are characterised by a persistently negative tuberculin skin test (TST) and/or persistently negative interferon-γ (IFN-γ) release assay (IGRA) and have been previously classified as 'resisters' [4]. Examining immunological factors associated with protection against latent TB infection or early clearance of *Mycobacterium tuberculosis* (*Mtb*) in these individuals has been viewed as an important step towards identifying protective immune correlates that could guide the development of new prevention strategies (novel TB vaccines) and curative measures for mitigating the global TB epidemic [6].

Although several studies have evaluated demographic, clinical and exposure risk factors associated with remaining TB uninfected despite high exposure [7–9], these epidemiological factors remain poorly understood. Among close household contacts of active TB cases, demographics and TB exposure characteristics such as younger age, increased number of household windows, fewer people living in the household [7] and BCG vaccination [9] have been associated with resistance to the

development of latent TB Infection [7,9]. On the other hand, one study in India did not show any epidemiological factor associated with a resister phenotype [8]. Defining the resister phenotype is further complicated by challenges associated with the lack of any gold standard to measure latent TB except for the TST or IGRAs, which do not fully capture spectrum before *Mtb* exposure [10]. Although these tests are indirect markers of *Mtb* exposure and potentially differ in sensitivity and specificity [11,12], the simultaneous use of both tests can increase the specificity for identifying those who are truly positive or negative [13].

Understating epidemiological factors associated with a resister phenotype will provide insight in identifying individuals who are truly resistant to TB infection rather than identifying those who are negative because of reduced exposure to TB. Moreover, this is also essential for accurately characterizing protective markers aimed to inform new prevention strategies. We identified HHCs of recently diagnosed pulmonary TB patients from South Africa that were TB infected using both the TST and IGRA. We evaluated the relationships between the demographics of index TB cases and HHCs, and environmental exposures at the household level with not having latent TB. We also used a modified epidemiologic risk score to determine further the risk of exposure and TB infection in HHCs [7].

## Methods

### Study population

We present a secondary analysis of a more extensive investigation to understand the role of immunological factors associated with remaining TB-uninfected among household contacts exposed to patients with infectious TB disease. This secondary analysis evaluates epidemiological factors associated with being TB uninfected despite exposure to an active TB case. To be eligible for inclusion in this study, index cases had to be ≥ 18 years of age, have rifampicin-sensitive TB based on the Xpert MTB/RIF assay®, and have at least one or more acid-fast bacilli (AFB) on sputum smear. Individuals who were rifampicin susceptible to pulmonary TB were asked to provide written informed consent to review their medical records and contact their household members for recruitment into the study. As per the South African national guidelines, all index cases were initiated on treatment by the Primary Health Care facilities [14].

Home visits to enrol household contacts were done within two to three weeks of the initial diagnosis of the TB index case by the study research teams comprised of professional nurses, research assistants, and quality control officers. The research teams first sought permission from the head of the household to recruit other household members. Permission to recruit HHCs was also sought if the head of the household was an index case. Household contacts ≥ 18 years of age who were available on the home visit were invited to participate. Research teams would return to the households to recruit HHCs who were not available on the visit day. Household contacts (HHC) were defined as individuals who had lived in the household of the TB index case for at least seven sequential days in the previous three months [15]. Those unavailable on the day were later contacted and invited to participate in the study.

Trained staff described the study individually to each HHC and obtained written informed consent before administering structured questionnaires. Information collected using the questionnaire include socio-demographics; contact with the index case; family relationship, sleeping relationship; time spent with index case; structure of the house (type of house, number of rooms/windows); duration of cough of the index case, and number of household members. Household structure information was also confirmed by inspection. HHCs also completed a TB symptom screening questionnaire; those who reported a productive cough, unintentional weight loss, fever, night sweats, dyspnoea, and pleuritic chest pain, were considered possible TB. These HHCs were excluded from the study and referred to local clinics for further testing and care. BCG vaccination status was confirmed by visualizing the BCG scar. Household contacts were also tested for HIV using finger-prick blood samples and the Alere Determine™ HIV1−2 rapid test (Alere, USA); samples with a positive result were retested using Uni-gold™ (Trinity Biotech, Ireland). Discordant results were resolved by an HIV-1 p24Ag ELISA test (HIV Combi PT, Roche Switzerland) performed on a venous blood sample. Household contacts who were HIV seropositive on

both rapid tests or by ELISA were referred for care and excluded from the study. Household contacts who reported having TB in the past or were currently undergoing TB treatment were also excluded from the study.

## TB risk assessment score

From the questionnaire, we generated an epidemiologic risk score for LTBI as previously reported [7]. The critical questions extracted from the questionnaire and medical records were whether (i) there was confirmation of pulmonary TB disease in the index case (all subjects in this trial were close contacts of definite pulmonary TB cases), (ii) the index case had smear-positive sputum, (iii) the index was currently coughing, (iv) the HHC was the spouse of the index case, (v) the HHC slept in the same bed as the index case, (vi) the HHC slept in the same room as the index case, (vii) the HHC lived with an index case in the same household, (viii) the HHC was the primary caregiver of the index case, (ix) the HHC saw the index case every day, and (x) whether there was more than one active TB case currently staying in the household. Each answer was scored as a one (Yes) and a zero (No) for a maximum total score of 10.

## TST and QFT testing

Venous blood samples were collected and tested using the QuantiFERON-TB Gold In-tube (QFT) assay per the manufacturer's instructions (QIAGEN, Hilden Germany). The QFT levels were estimated using an ELISA reader and expressed as interferon-γ concentrations in international units per ml (IU/ml). A QFT test result ≥0.35 IU/ml (after background subtraction of the TB NIL tube) was considered positive. A TST was then administered using the Mantoux method by injecting a 0.1 mL of liquid containing two units of tuberculin purified protein derivative PPD-RT23 2TU PPD (purified protein derivative) (SepSci, South Africa) into the epidermis of the forearm. The clinician returned to the household between 48–72 hours to measure palpable induration diameter (mm). TST induration ≥ 5 mm was considered positive. HIV-uninfected household contacts who tested positive on both QFT and TST (QFT + TST+) were defined as having LTBI (cases), and those who tested negative (QFT-TST-) were defined as TB uninfected. The remaining HHCs had discordant results between QFT and TST (i.e., QFT + TST-/QFT-TST+) and could not be ascertained as having or not having LTBI and therefore were excluded from the analysis. LTBI and TB uninfected HHCs had a repeat QFT- and/or TST- test at three months follow up, and those who tested negative were defined as persistently TB uninfected HHCs.

## Data analysis

Baseline characteristics were described using frequencies and percentages, and predictors related to the outcome variable (QFT-TST-) were also described using frequencies and percentages. For the inferential statistics, we assess possible significant clustering due to the sampling distribution by first fitting an intercept-only mixed effect model to determine a significant improvement in fit relative to the standard binary logistic regression. However, with a chi-square ($\chi^2$) test of p-value of 0.4 for no difference in the fit between models, coupled with the primary objective of identifying significant predictors associated with remaining QFT-TST- at baseline and at three months follow-up despite TB exposure. Univariable analysis was used to evaluate the independent association between each predictor and the outcome (QFT-TST- at baseline or QFT-/QFT- at three months). All variables with a threshold p-value ≤ 0.10 in univariate analysis were retained in a final multivariable logistic model. While the binary logistic regression was used to evaluate the significant association of each predictor in the context of others, we also developed a risk score from 10 variables considered to increase the risk of transmission due proximity and exposure to the index case [7]. The highest possible risk score was 10; HHCs with ≥6 risk score were categorized as having a high risk for LTBI [7,16]. We adjusted for the risk score in the multivariable analysis while excluding the endogenous variables of the risk score to avoid multicollinearity and stratified the analysis by HHCs with high and low-risk scores. Results are reported as crude, and adjusted odds ratios (aOR) with corresponding 95% confidence interval (CI) and p-values. The data analysis was done using STATA V16.0 (Stata Corporation, College Station, Texas, USA).

## Ethical consideration

The parent study was reviewed and approved by the University of Witwatersrand Human Research Ethics Committee, and the Research Committee of the Northwest Provincial Department of Health. All participants provided written informed consent prior to inclusion.

## Results

### Index cases characteristics

Between April 2015 and April 2017, a total of 166 adults with active pulmonary TB disease (index cases) were recruited at 24 local primary health care facilities in Rustenburg, South Africa. Of the 166 index cases screened, 151 (90.1%) were enrolled in the study. Index cases that were excluded included: five index's household contacts refused to take part in the study, seven of the index's household contacts had known HIV-positive status prior to screening, one had a negative sputum smear result, and two were unavailable to participate in the study (Fig 1). Of the 151 enrolled index cases, 92 (60.9%) were male, nearly half had a positive sputum grade of 3+ (74,49.0%) indicating high mycobacterial loads, and a majority had a duration of cough lasting between 5–10 weeks (67,60.9%) (Table 1). More than two-thirds of index cases (74,67.3%) had only one household contact, while others (36,32.7%) had two or more household contacts evaluated for risk for TB infection. Most index cases dwelled in brick houses (108,72%) and homes with more than three habitable rooms (97,64.7%).

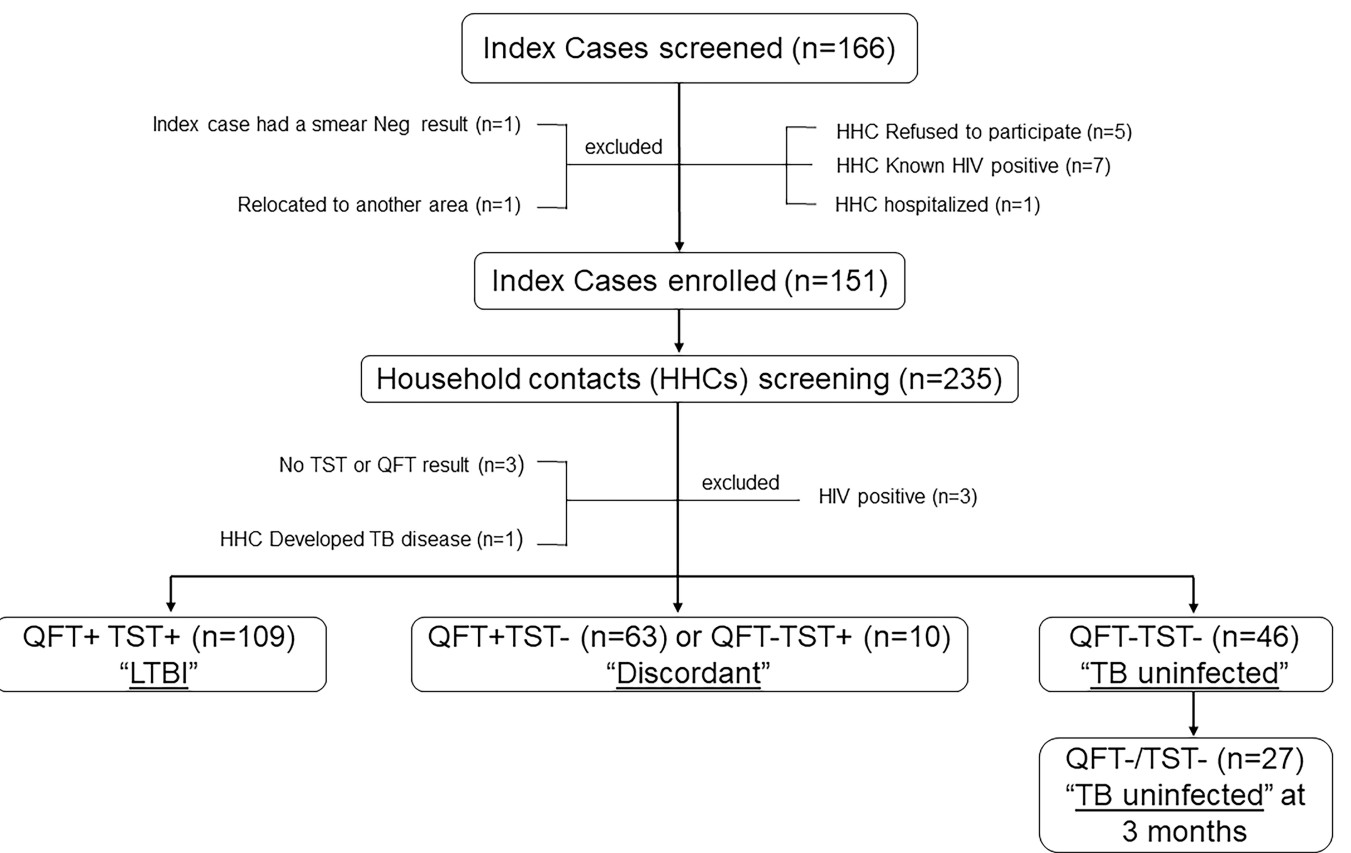

**Fig 1. Screening and enrolment of TB index cases and their household contacts (HC), Rustenburg, South Africa.**

**Table 1. Baseline characteristics of pulmonary TB index cases, and household contacts, Rustenburg, South Africa.**

| Baseline Characteristics | Index Cases (N=151) | Household Contacts (N=228) |
|---|---|---|
| **Age** | N (%) | N (%) |
| 18-30 | 57(37.8) | 114(50.0) |
| 31-40 | 34(22.5) | 26(11.4) |
| >40 | 60(39.7) | 88(38.6) |
| **Gender** | | |
| Female | 59(39.1) | 88(38.6) |
| Male | 92(60.9) | 140(61.4) |
| **Sputum smear grading** | | |
| 1+ | 44(29.1) | |
| 2+ | 33(21.9) | |
| 3+ | 74(49.0) | |
| **Duration of cough (in weeks)** | | |
| 1-4 | 25(22.7) | |
| 5-10 | 67(60.9) | |
| >10 | 18(16.4) | |
| **Household contacts per index case** | | |
| 1 | 74(67.3) | |
| 2 | 28(25.5) | |
| ≥3 | 8(7.3) | |
| **Presence of BCG scar** | | |
| No | 88(80.0) | 146(64) |
| Yes | 22(20.0) | 46(20.2) |
| Uncertain | | 36(15.8) |
| **Relationship to index case** | | |
| Spouse | | 26(11.4) |
| Child | | 55(24.1) |
| Other relative | | 137(60.1) |
| Non-relative | | 10(4.4) |
| **Time spent with index, hrs/day** | | |
| 2-12 | | 92(40.4) |
| ≥13 | | 136(59.7) |
| **Proximity** | | |
| Same room, same bed | | 59(25.9) |
| Same room, different bed | | 124(54.4) |
| Same house, different room | | 14(6.1) |
| Same Yard | | 31(13.6) |
| **Number of household occupants** | | |
| 1-5 | 95(63.3) | 127(55.7) |
| 6-10 | 43(28.7) | 79(34.7) |
| >10 | 12(8.0) | 22(9.7) |
| **Structure of house** | | |
| Shack | 42(28) | 65(28.5) |
| RDP/Brick | 108(72) | 163(71.5) |
| **Number of house windows** | | |
| 0-3 | 51(34.0) | 74(32.5) |
| 4-6 | 54(36.0) | 76(33.3) |

*(Continued)*

**Table 1.** (Continued)

| Baseline Characteristics | Index Cases (N=151) | Household Contacts (N=228) |
|---|---|---|
| ≥7 | 45(30.0) | 78(34.2) |
| **Number of habitable rooms** | | |
| 1-3 | 53(35.3) | 78(34.2) |
| >3 | 97(64.7) | 150(65.8) |
| **Baseline Characteristics** | **Index Cases (N=151)** | **Household Contacts (N=228)** |
| **Age** | **N (%)** | **N (%)** |
| 18-30 | 57(37.8) | 114(50) |
| 31-40 | 34(22.5) | 26(11.4) |
| >40 | 60(39.7) | 88(38.6) |
| **Gender** | | |
| Female | 59(39.1) | 88(38.6) |
| Male | 92(60.9) | 140(61.4) |
| **Sputum smear grading** | | |
| 1+ | 44(29.1) | |
| 2+ | 33(21.9) | |
| 3+ | 74(49) | |
| **Duration of cough (in weeks)** | | |
| 1-4 | 25(16.6) | |
| 5-10 | 67(44.4) | |
| >10 | 18(11.9) | |
| Missing | 41(27.2) | |
| **Household contacts per index case** | | |
| **1** | 74(49) | |
| **2** | 28(18.5) | |
| **≥3** | 8(5.3) | |
| Missing | 41(27.2) | |
| **Presence of BCG scar** | | |
| No | 88(58.3) | 146 (64.1) |
| Yes | 22(14.6) | 46 (20.18) |
| Uncertain | | 36 (15.79) |
| Missing | 41(27.2) | |
| **Relationship to the index case** | | |
| Spouse | | 26 (11.4) |
| Child | | 55 (24.1) |
| Other/Non-relative | | 147 (64.5) |
| **Time spent with index, hrs/day** | | |
| 2-12 | | 92(40.4) |
| ≥13 | | 136(59.7) |
| **Proximity** | | |
| Same room, same bed | | 59(25.9) |
| Same room, different bed | | 124(54.4) |
| Same house, different room | | 14(6.1) |
| Same Yard | | 31(13.6) |
| **Number of household occupants** | | |
| 1-5 | 95(63.3) | 127(55.7) |

*(Continued)*

**Table 1.** (Continued)

| Baseline Characteristics | Index Cases (N=151) | Household Contacts (N=228) |
|---|---|---|
| 6-10 | 43(28.7) | 79(34.7) |
| >10 | 12(8.0) | 22(9.7) |
| **Structure of house** | | |
| Shack | 42(28.0) | 65(28.5) |
| RDP/Brick | 108(72.0) | 163(71.5) |
| **Number of house windows** | | |
| 0-3 | 51(34.0) | 74(32.5) |
| 4-6 | 54(36.0) | 76(33.3) |
| ≥7 | 45(30.0) | 78(34.2) |
| **Number of habitable rooms** | | |
| 1-3 | 53(35.3) | 78(34.2) |
| >3 | 97(64.7) | 150(65.8) |

## Household contacts baseline characteristics

We screened 235 household contacts, of whom 228 (97.0%) participated. Of these, 109 (46.4%) were found to be QFT+TST+ and 46 (19.5%) were QFT-TST- (Fig 1). A total of 73 (31%) household contacts had discordant QFT/TST results; 63 (26.6%) were QFT+TST- and 10 (4.3%) were QFT-TST+. Of the 46 QFT-TST- HHCs who were negative at baseline, over half (58.6%) remained persistently QFT-/TST- negative after three months. Three (1.3%) household contacts had missing QFT/TST results; the other three HHCs were found to be HIV infected, of whom one had QFT+TST+ and two had QFT/TST discordant results. One household contact had symptoms of TB during screening and was subsequently diagnosed with active TB disease. Baseline characteristics of the 228 household contacts are shown in Table 1. Half of the household contacts were between 18 and 30 years old; predominantly female (61.4%), and the majority (64.1%, 146 of 228) had no visible BCG scar. A total of 147 (64.5%) reported that they were next of kin or relatives of the index case and 24.2% were children of the index cases, and about 60% of the household contacts spent more than 13 hours per day with the index case. Nearly three-quarters (71.5%) of household contacts stayed in brick houses, with a majority (65.8%) having more than three habitable rooms.

## Comparisons of QFT+TST+ and QFT-TST- household contacts at baseline

The QFT+TST+ household contacts of TB index patients tended to be older, with 37.6% aged > 40 years versus 23.9% of QFT-TST- HHCs (Table 2). The proportion of QFT+TST+ and QFT-TST- HHCs had a similar distribution of sex, the presence of a BCG scar, sleep distance between household contacts and index case, time spent with index case as well as the epidemiologic risk score (i.e., similar proportion of "higher" vs. "lower" risk, per a previously designed tool [7]). We also examined clinical characteristics of the TB index cases, and the index cases of QFT-TST- HHCs were less likely to report a 5–10-week long-term cough compared to the QFT+TST+ (47.8% versus 67.9%). However, we found similar distribution in the severity of TB disease (based on sputum smear grade) of the index case between the two study groups. A higher proportion of QFT-TST- HHCs tended to stay in homes with seven or more windows than QFT+TST+ contacts (45.7% versus 28.4%) and more than three habitable rooms (84.8% versus 55.0%).

## Factors associated with being QFT-TST- or persistently QFT-/TST-

Next, we used logistic regression analysis to identify factors independently associated with TST-QFT- at baseline (Table 3). The odds of being QFT-TST- increased with age ≤ 30 (OR: 2.42, 95%CI: 1.1–5.29); and in HHCs residing in houses

**Table 2. Results of QuantiFERON TB (QFT) and tuberculin skins tests (TST) among household contacts of TB index cases at baseline.**

| Index contact characteristics | QFT+/TST+ N (%) | QFT-/TST- N (%) | Total N (%) |
|---|---|---|---|
| **Age** | | | |
| 18-30 | 48(44) | 29(63) | 77(49.7) |
| 31-40 | 20(18.4) | 6(13) | 26(16.8) |
| >40 | 41(37.6) | 11(23.9) | 52(33.6) |
| **Gender** | | | |
| Female | 69(63.3) | 29(63.0) | 98(63.3) |
| Male | 40(36.7) | 17(37.0) | 57(36.8) |
| **Presence of BCG scar** | | | |
| No | 73(67.0) | 27(58.7) | 100(64.6) |
| Yes | 18(16.5) | 13(28.3) | 31(20.0) |
| Uncertain | 18(16.5) | 6(13.0) | 24(15.5) |
| **Relationship to the index case** | | | |
| Spouse | 14(12.8) | 1(2.2) | 15(9.7) |
| Child | 30(27.5) | 9(19.6) | 39(25.2) |
| Other/Non-relative | 65(59.6) | 36(78.3) | 101(65.2) |
| **Time spent with index, hrs/day** | | | |
| 2-12 | 45(41.3) | 15(32.6) | 60(38.8) |
| ≥13 | 64(58.7) | 31(67.4) | 95(61.3) |
| **Proximity** | | | |
| Same room, same bed | 28(25.7) | 12(26.1) | 40(25.9) |
| Same room, different bed | 58(53.2) | 30(65.2) | 88(56.8) |
| Same house, different room | 7(6.4) | 1(2.2) | 8(5.2) |
| Same Yard | 16(14.7) | 3(6.5) | 19(12.3) |
| **Number of household occupants** | | | |
| 1-5 | 64(58.7) | 21(45.7) | 85(54.9) |
| 6-10 | 38(34.9) | 18(39.1) | 56(36.2) |
| >10 | 7(6.4) | 7(15.2) | 14(9.1) |
| **Risk Score** | | | |
| 0-5 | 44(40.4) | 20(43.5) | 64(41.3) |
| 6-10 | 65(59.6) | 26(56.5) | 91(58.8) |
| **Index Case Characteristics** | | | |
| **Sputum smear grading** | | | |
| 1+ | 32(29.36) | 14(30.4) | 46(29.7) |
| 2+ | 28(25.69) | 12(26.1) | 40(25.9) |
| 3+ | 49(44.95) | 20(43.5) | 69(44.6) |
| **Duration of cough (in weeks)** | | | |
| 1-4 | 18(16.5) | 12(26.1) | 30(19.4) |
| 5-10 | 74(67.9) | 22(47.8) | 96(62.0) |
| >10 | 17(15.6) | 12(26.1) | 29(18.8) |
| **Household contacts per index case** | | | |
| 1 | 50(45.9) | 24(52.2) | 74(47.75) |
| 2 | 43(39.5) | 13(28.3) | 56(36.13) |
| ≥3 | 16(14.7) | 9(19.6) | 25(16.13) |
| **Index Case housing characteristic** | | | |
| **Structure of house** | | | |

*(Continued)*

**Table 2.** (Continued)

| Index contact characteristics | QFT+/TST+<br>N (%) | QFT-/TST-<br>N (%) | Total<br>N (%) |
|---|---|---|---|
| Shack | 39(35.8) | 11(23.9) | 50(32.3) |
| Brick | 70(64.2) | 35(76.1) | 105(67.8) |
| **Number of house windows** | | | |
| 0-3 | 46(42.2) | 9(19.6) | 55(35.5) |
| 4-6 | 32(29.4) | 16(34.8) | 48(31) |
| ≥7 | 31(28.4) | 21(45.7) | 52(33.6) |
| **Number of habitable rooms** | | | |
| 1-3 | 49(45) | 7(15.2) | 56(36.2) |
| >3 | 60(55.1) | 39(84.8) | 99(63.9) |

Chi-squared and Fischer Exact were used for statistical comparisons.

with seven or more windows (OR: 3.47, 95% CI: 1.41–8.56) and more than three habitable rooms (OR:4.55, 95% CI: 1.88–11.07), and being next of kin or relative of the index case (OR: 2.44, 95% CI: 1.09–5.41). Factors such as gender, presence of a BCG scar, index case with sputum smear grade, type of house, spouse/child and the epidemiolocal risk score were not significantly associated with QFT-TST- status.

After controlling for gender and prior TB exposure in a multivariable analysis, younger age (aOR: 2.81, 95%CI: 1.23–6.47) and increased number of habitable rooms (aOR:4.62, 95%CI: 1.81–11.79) remained significantly associated with QFT-TST-. A similar trend was also observed when three months of follow-up were used to identify those with persistent QFT+TST+ and QFT-TST- status; younger age (aOR:4.24, 95%CI: 1.30–13.83) and increased number of habitable rooms (aOR:7.08, 95%CI: 1.67–30.0) both remained significantly associated with being QFT-/TST- (Table 4). When we refined our analysis by stratifying HHCs with high (≥ 6) and low (< 6) epidemiological risk scores, the increased number of habitable rooms was found to be associated with QFT-TST- status only in HHCs with high-risk scores (aOR: 7.91, 95% CI: 2.16–28.94) (S1 Table). On the other hand, the odds of being TST-QFT- were increased in HHCs whose age was ≤ 30 (aOR: 9.34, 95% CI: 1.12–78.16) among those with low-risk scores (S1 Table).

## Discussion

Increasing evidence suggests that a small fraction of individuals may naturally resist TB infection despite high exposure to an infectious TB case [6,8,15,17]. We conducted a cross-sectional and longitudinal evaluation of factors associated with being TB uninfected among household contacts exposed to TB and found that at baseline about 20% of household contacts of active pulmonary TB cases were TB uninfected based on negative QFT and TST tests. Over half (58.6%) of the TB uninfected HHCs remained persistently QFT/TST negative at three months follow-up. The overall prevalence of about 20% reported in our study is higher than those of HHCs reported in India and Uganda. The differences in prevalence may be explained by the criteria we used to define TB-exposed uninfected HHCs (TST<5 mm and QFT<0.35 IU/ml). A study in India reported a prevalence among HHCs of 7% exposed TB-uninfected when using readout of <5mm TST induration as a cut-off [8]. Another study conducted in Uganda also reported a prevalence of 12% when using a less stringent criteria of <10mm TST induration [7]. Not surprisingly, it has been reported that the prevalence of exposed TB-uninfected HHCs ranges between 5–50% depending on the criteria used for a TST positive result or the use of either a TST or QFT result and both [4,18]. While the sensitivity and specificity of TST or QFT results determines the likelihood of someone having a resistor phenotype, epidemiological factors such as Mycobacterial lineage and infectivity of the TB index case, proximity to the index case, intensity, and duration of exposure provide insights to discern those with a resistor phenotype accurately.

In line with previous findings [7], we found that exposed TB-uninfected HHCs were likely to be younger (18–30 years of age), compared to HHCs with LTBI. Increasing age has been previously linked with increased risk for LTBI in various

**Table 3. Factors associated with being QFT-TST- amongst household contacts of smear-positive pulmonary TB index cases (n = 46) at baseline.**

| Variable | Total | QFT-/TST- | | Univariate analysis | | Multivariate analysis | |
|---|---|---|---|---|---|---|---|
| | N | n | % | Unadjusted OR (95% CI) | p-value | Adjusted OR (95% CI) | p-value |
| **Age, years** | | | | | | | |
| 18-30 | 77 | 29 | **0.38** | 2.42 (1.11-5.29) | **0.03** | 2.81 (1.23-6.47) | **0.02** |
| 31-40 | 26 | 6 | **0.23** | 1.54 (0.46-5.16) | 0.49 | 1.53 (0.43-5.51) | 0.53 |
| >40 | 52 | 11 | **0.21** | Ref. | | | |
| **Gender** | | | | | | | |
| Female | 98 | 29 | **29.6** | Ref. | | | |
| Male | 57 | 17 | **29.8** | 0.99 (0.49-2.02) | 0.98 | | |
| **Presence of BCG Scar** | | | | | | | |
| No | 100 | 27 | **27.0** | Ref. | | | |
| Yes | 31 | 13 | **41.9** | 1.96 (0.85-4.52) | 0.12 | | |
| **Duration of cough of index, wks.** | | | | | | | |
| 1-4 | 30 | 12 | **40.0** | 2.25 (0.94-5.37) | 0.07 | 2.3 (0.90-5.90) | 0.09 |
| 5-10 | 96 | 21 | **22.9** | Ref | | | |
| >10 | 29 | 12 | **41.4** | 1.06 (0.38-3) | 0.92 | 1.85 (0.71-4.79) | 0.22 |
| **Time Spent with index, hrs.** | | | | | | | |
| 2-12 | 60 | 15 | **25.0** | Ref. | | | |
| ≥13 | 95 | 31 | **32.6** | 1.46 (0.71-3.0) | 0.32 | | |
| **Number of household occupants** | | | | | | | |
| 1-5 | 85 | 21 | **0.25** | Ref. | | | |
| 6-10 | 56 | 18 | **0.32** | 1.45 (0.69-3.05) | 0.34 | | |
| >10 | 14 | 7 | **0.50** | 3.05 (0.96-9.71) | 0.06 | | |
| **Household structure** | | | | | | | |
| Shack | 50 | 11 | **22.0** | Ref. | | | |
| Brick | 105 | 35 | **33.3** | 1.78 (0.82-3.88) | 0.16 | | |
| **Number of house windows** | | | | | | | |
| 0-3 | 55 | 9 | **16.4** | Ref. | | | |
| 4-6 | 48 | 16 | **33.3** | 2.56 (1.01-6.50) | 0.05 | | |
| ≥7 | 52 | 21 | **40.4** | 3.47 (1.41-8.56) | **0.01** | | |
| **Household habitable rooms** | | | | | | | |
| 1-3 | 56 | 7 | **12.5** | Ref. | | Ref. | |
| >3 | 99 | 39 | **39.4** | 4.55 (1.88-11.07) | **0.01** | 4.62 (1.81-11.79) | **0.001** |
| **Relationship to the index case** | | | | | | | |
| Spouse/child | 54 | 10 | **0.19** | Ref. | Ref | | |
| Others/Non-relative | 101 | 36 | **0.36** | 2.44 (1.09-5.41) | **0.03** | | |
| **Proximity** | | | | | | | |
| Same room, same bed | 40 | 12 | **0.30** | Ref | | | |
| Same room, different bed | 88 | 30 | **0.34** | 1.21 (0.54-2.71) | 0.65 | | |
| Same house, different room/Same yard | 27 | 4 | **0.15** | 0.41 (0.12-1.43) | 0.16 | | |
| **Risk score** | | | | | | | |
| Low risk | 64 | 20 | **31.0** | | | | |
| High Risk | 91 | 26 | **29.0** | 0.88 (0.44-1.77) | 0.72 | | |

Reconstruction and Development Program (RDP). All variables in this model were included without predefined categorization as independent or adjustment variables. As such, the reported estimates should not be interpreted as causal effects but rather as statistical associations that warrant further investigation.

**Table 4. Factors associated with being QFT-/TST- amongst household contacts of smear-positive pulmonary TB index cases (n=27) at 3 months.**

| Variable | Total | QFT-/TST- | | Univariate analysis | | Multivariate analysis | |
|---|---|---|---|---|---|---|---|
| | N | n | % | Unadjusted OR (95% CI) | p-value | Adjusted OR (95% CI) | p-value |
| **Age, years** | | | | | | | |
| 18-30 | 37 | 20 | 0.54 | 4.24(1.3-13.83) | 0.02 | 4.24(1.3-13.83) | 0.02 |
| 31-40 | 6 | 2 | 0.33 | 7.2(0.54-96.64) | 0.14 | 7.2(0.54-96.64) | 0.14 |
| >40 | 20 | 5 | 0.25 | Ref. | | | |
| **Gender** | | | | | | | |
| Female | 22 | 10 | 0.45 | | | | |
| Male | 41 | 17 | 0.41 | 0.85(0.3-2.42) | 0.76 | | |
| **Presence of BCG Scar** | | | | | | | |
| No | 49 | 21 | 0.43 | Ref. | | | |
| Yes | 14 | 6 | 0.43 | 1(0.31-3.33) | 1 | | |
| **Duration of cough of index, wks.** | | | | | | | |
| 1-4 | 13 | 8 | 0.62 | 2.91(0.77-11.09) | 0.12 | | |
| 5-10 | 31 | 11 | 0.35 | Ref | | | |
| >10 | 19 | 8 | 0.42 | 1.33(0.42-4.27) | 0.64 | | |
| **Time Spent with index, hrs.** | | | | | | | |
| 2-12 | 23 | 9 | 0.39 | Ref. | | | |
| ≥13 | 40 | 18 | 0.45 | 1.28(0.45-3.62) | 0.66 | | |
| **Number of household occupants** | | | | | | | |
| 1-5 | 38 | 10 | 0.26 | Ref. | | | |
| 6-10 | 18 | 13 | 0.72 | 7.28(2.07-25.64) | 0.01 | | |
| >10 | 7 | 4 | 0.57 | 3.74(0.71-19.68) | 0.12 | | |
| **Household structure** | | | | | | | |
| Shack | 24 | 8 | 0.33 | Ref. | | | |
| RDP/Brick | 39 | 19 | 0.49 | 1.9(0.67-5.47) | 0.24 | | |
| **Number of house windows** | | | | | | | |
| 0-3 | 23 | 6 | 0.26 | Ref. | | | |
| 4-6 | 20 | 11 | 0.55 | 3.47(0.97-12.48) | 0.06 | | |
| ≥7 | 20 | 10 | 0.50 | 2.84(0.79-10.18) | 0.11 | | |
| **Household habitable rooms** | | | | | | | |
| 1-3 | 19 | 3 | 0.16 | Ref. | | Ref. | |
| >3 | 44 | 24 | 0.55 | 6.4(1.63-25.15) | 0.01 | 7.08(1.67-30) | 0.01 |
| **Relationship to index case** | | | | | | | |
| Spouse/child | 21 | 6 | 0.29 | Ref. | | | |
| Other relative/Nonrelative | 42 | 21 | 0.50 | 2.5(0.82-7.69) | 0.11 | | |
| **Proximity** | | | | | | | |
| Same room, same bed | 16 | 6 | 0.38 | Ref | | | |
| Same room different bed/same house different room | 47 | 21 | 0.45 | 1.35(0.43-4.32) | 0.62 | | |
| **Risk Score** | | | | | | | |
| Low Risk | 27 | 13 | 0.48 | | | | |
| High Risk | 36 | 14 | 0.39 | 0.69(0.25-1.89) | 0.47 | | |

All variables in this model were included without predefined categorization as independent or adjustment variables. As such, the reported estimates should not be interpreted as causal effects but rather as statistical associations that warrant further investigation.

endemic settings [19–22]. Furthermore, compared to TB exposed uninfected household contacts, a larger proportion of LTBI HHCs lived with index TB cases who reported a longer duration of cough (5–10-weeks), and stayed in homes that had relatively few windows. Together, these findings may suggest less overall exposure among exposed uninfected contacts.

We also found that TB uninfected HHCs were more likely to stay in homes that had a greater number of habitable rooms, further highlighting a lower exposure risk to *Mtb* among this group. However, level of exposure and incident infection also depends on accumulation of multiple risk factors such as proximity to the index case, intensity and/or duration of exposure as well as clinical factors of the index case [5,7]. In this study, the distribution of other risk factors were comparable between TB infected and uninfected household contacts. Moreover, similar to previous studies [7,8], the TB risk score consisting of multiple TB infection risk factors was found to be similar between the HHCs groups, and arguing for similar exposure levels. However, when we stratified the analyses based on high and low risk scores (S1 Table), we found that the number of household rooms significantly associated with QFT-TST- status among HHCs with high-risk scores, further suggesting a potential reduced TB exposure among QFT-TST- HHCs.

Our study has several limitations. A major limitation for this study is that our comparisons are based on a small sample size from one site in Rustenburg, which makes it difficult to generalise these findings to other contexts. Additionally, we did not include variables assessing educational level, financial income, and level of awareness of TB on the questionnaire, which could potentially act as confounders. We also did not use chest X-ray and sputum culture to quantify the extent of TB disease and level of infectiousness among the index cases. However, use of AFB smear allowed us to estimate level of bacillary burden on sputum as well indicating that index cases enrolled in the study were infectious.

In summary, our findings demonstrate that the prevalence of QFT-TST- individuals at the time of investigation around TB index cases was approximately 20% in our cohort. Many participants in our study with negative test results may also have had lower rates of exposure; teasing out low exposure risk from a true 'resistor' phenotype remains a challenge. We also failed to identify a groups with a commonly used TB risk calculator, which is likely due to the small number of participants examined in this study. On the other hand, recent studies using larger cohorts failed to demonstrate meaningful epidemiological or clinical factors associated remaining TB uninfected among exposed household contacts [8,18]. Additional studies are needed to fully elucidate epidemiological factors as well as immunological and/or genetic biomarkers that confer resistance to *Mtb* infection in well-defined cohorts.

## Supporting information

**S1 Table. Factors associated with being QFT-TST- amongst household contacts stratified by risk score.**
(DOCX)

## Acknowledgments

We would like to acknowledge participants for their generous participation in this study. We thank The Aurum Institute Rustenburg Clinical Research Site for implementation of the parent study.

## Author contributions

**Conceptualization:** Daniel Hoft, Salome Charalambous, Gavin Churchyard, Pholo Maenetje.

**Formal analysis:** Matthew Oladimeji, Mandla Mlotshwa.

**Investigation:** Pholo Maenetje.

**Methodology:** Pholo Maenetje.

**Project administration:** Pholo Maenetje.

**Supervision:** Gavin Churchyard.

**Writing – original draft:** Nomfanelo Maenetje, Pholo Maenetje.

**Writing – review & editing:** Nomfanelo Maenetje, Matthew Oladimeji, Mandla Mlotshwa, Daniel Hoft, Christina Lindan, Robert Wallis, Salome Charalambous, Vinodh Edward, Andrew Fiore-Gartland, Jerry Shai, Pholo Maenetje.

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
