## [Decision Letter · Decision Letter 0]

17 Jan 2025

PONE-D-24-20846Factors associated with remaining TB-uninfected despite household contact with active TB patients in South AfricaPLOS ONE

Dear Dr. Maenetje,

Thank you for submitting your manuscript to PLOS ONE. After careful consideration, we feel that it has merit but does not fully meet PLOS ONE’s publication criteria as it currently stands. Therefore, we invite you to submit a revised version of the manuscript that addresses the points raised during the review process.

We look forward to receiving your revised manuscript.

Kind regards,

Xiangwei Li

Academic Editor

PLOS ONE

Journal Requirements:

Reviewers' comments:

Reviewer's Responses to Questions

**Comments to the Author**

1. Is the manuscript technically sound, and do the data support the conclusions?

Reviewer #1: Yes

Reviewer #2: Yes

Reviewer #3: Partly

2. Has the statistical analysis been performed appropriately and rigorously? 

Reviewer #1: Yes

Reviewer #2: Yes

Reviewer #3: No

3. Have the authors made all data underlying the findings in their manuscript fully available?

Reviewer #1: Yes

Reviewer #2: Yes

Reviewer #3: Yes

4. Is the manuscript presented in an intelligible fashion and written in standard English?

Reviewer #1: Yes

Reviewer #2: Yes

Reviewer #3: Yes

5. Review Comments to the Author

Reviewer #1: I read the manuscript with great interest. The idea and the aim of the study, although not original, are very relevant and important.

The work has been carried out and described very carefully. The methods, results, and conclusions are appropriate.

However, the fundamental question of whether some individuals are resistant to tuberculosis, or whether contact with active pulmonary tuberculosis was simply shorter in uninfected individuals, remains unanswered after this study.

Perhaps the authors still have the opportunity to compare infected and uninfected groups in terms of educational level, financial income, and level of awareness of TB. I hypothesize that higher education, better family financial income, and higher awareness may have led to less contact with a person with active TB, and better guarding or use of personal protective clue.

Reviewer #2: I understand the challenges of using a questionnaire to measure exposure but the team used a reasonable approach, I do have some concern that the sample size was small but they addressed this in the study limitations and with this small sample size where able to show interesting findings

Reviewer #3: 1. In view of lines 112 and 113 which reads ‘immunological factors associated with remaining free of TB infection and the timing of the variables that were analysed in the tables (results), the following title is suggested:

“Epidemiological factors associated with immunological resistance in household contacts exposed to active tuberculosis in South Africa: an analysis using multivariable regression”

2. In the multivariate data analysis, a logistic regression Why [if the prevalence of the variable of interest was greater than 10%] was applied, and it is recommended that the authors justify the selection of this model rather than a Poisson regression with robust variance, as this explanation will provide key information for the reader. For a useful reference on this methodological decision.

See:

Thompson ML, Myers JE, Kriebel D. Prevalence odds ratio or prevalence ratio in the analysis of cross sectional data: what is to be done? Occup Environ Med. 1998 Apr;55(4):272-7. doi: 10.1136/oem.55.4.272.

McNutt LA, Wu C, Xue X, Hafner JP. Estimación del riesgo relativo en estudios de cohortes y ensayos clínicos de resultados comunes. Am J Epidemiol. 2003 May 15;157(10):940-3. doi: 10.1093/aje/kwg074.

3. I suggest including in the methodology section a clear specification of ‘VARIABLES’: independent variable, dependent variable, and adjustment or confounding variables. In addition, it would be advisable to add a footnote to the corresponding table, indicating that certain variables are not interpretable and including a brief justification. This will provide greater clarity to the reader and reduce the risk of misinterpretation regarding the role of the adjustment variables.

You may find it useful to consult the following reference:

Akinkugbe AA, Simon AM, Brody ER. A scoping review of Table 2 fallacy in the oral health literature. Community Dent Oral Epidemiol. 2021 Apr;49(2):103-109. doi: 10.1111/cdoe.12617.

6. PLOS authors have the option to publish the peer review history of their article (what does this mean? ). If published, this will include your full peer review and any attached files.

**Do you want your identity to be public for this peer review?** For information about this choice, including consent withdrawal, please see our Privacy Policy .

Reviewer #1: No

Reviewer #2: No

Reviewer #3: **Yes: ** Huamani-Echaccaya Jose Luis

---

## [Author Response · Author response to Decision Letter 1]

6 Feb 2025

Editors Comments

1. Please ensure that your manuscript meets PLOS ONE's style requirements, including those for file naming. The PLOS ONE style templates can be found at:

• We have formatted the manuscript according to the provided guidelines

• We have removed the funding related information from the manuscript and updated the Funding Statement on the cover letter.

3. We note that the grant information you provided in the ‘Funding Information’ and ‘Financial Disclosure’ sections do not match

• We have now matched the Funding statement and ‘Financial Disclosure’

4. When completing the data availability statement of the submission form, you indicated that you will make your data available on acceptance. We strongly recommend all authors decide on a data sharing plan before acceptance, as the process can be lengthy and hold up publication timelines. Please note that, though access restrictions are acceptable now, your entire data will need to be made freely accessible if your manuscript is accepted for publication. This policy applies to all data except where public deposition would breach compliance with the protocol approved by your research ethics board. If you are unable to adhere to our open data policy, please kindly revise your statement to explain your reasoning and we will seek the editor's input on an exemption. Please be assured that, once you have provided your new statement, the assessment of your exemption will not hold up the peer review process

• This is noted

• The corresponding author now has ORCID iD linked to this submission.

• We have provided full details on the ethics committee

Reviewers Comments

Reviewer #1:

1. I read the manuscript with great interest. The idea and the aim of the study, although not original, are very relevant and important. The work has been carried out and described very carefully. The methods, results, and conclusions are appropriate. However, the fundamental question of whether some individuals are resistant to tuberculosis, or whether contact with active pulmonary tuberculosis was simply shorter in uninfected individuals, remains unanswered after this study.

Perhaps the authors still have the opportunity to compare infected and uninfected groups in terms of educational level, financial income, and level of awareness of TB. I hypothesize that higher education, better family financial income, and higher awareness may have led to less contact with a person with active TB, and better guarding or use of personal protective clue.

• We acknowledge that factors such as educational level, financial income, and level of awareness of TB could also be potential confounders, however, we modelled our questionnaires based on previous similar epidemiological and clinical studies such as Ma et al [7], Mave et al [8] and Verrall et al [9]. Unfortunately, we did not collect data on education level, better family financial income and TB awareness. However, we used household structure (RDP/Brick house), number of house windows and number of household habitable rooms as proxy for family financial income. We have acknowledged this as a study limitation and added text to the discussion section accordingly (Lines 356- 358).

Reviewer #2:

1. I understand the challenges of using a questionnaire to measure exposure but the team used a reasonable approach, I do have some concern that the sample size was small but they addressed this in the study limitations and with this small sample size where able to show interesting findings

• Thank you for acknowledging our approach to using the questionnaire to measure exposure and for recognizing the efforts we made to address the limitations associated with the small sample size. We agree that the sample size was a limitation, as noted in the manuscript. Despite this limitation, we were able to uncover interesting and meaningful findings. To further strengthen the manuscript, we have revisited the discussion section to ensure the implications of our findings are clearly articulated and contextualized within the study's constraints. We have also emphasized that while the sample size was small, our results contribute valuable insights into the field and provide a foundation for future research with larger cohorts. Thank you for your thoughtful feedback, which has helped us improve the manuscript.

Reviewer #3:

1. In view of lines 112 and 113 which reads ‘immunological factors associated with remaining free of TB infection and the timing of the variables that were analysed in the tables (results), the following title is suggested:

“Epidemiological factors associated with immunological resistance in household contacts exposed to active tuberculosis in South Africa: an analysis using multivariable regression”

• We thank the reviewer for this comment and have updated the manuscript title requested.

2. In the multivariate data analysis, a logistic regression Why [if the prevalence of the variable of interest was greater than 10%] was applied, and it is recommended that the authors justify the selection of this model rather than a Poisson regression with robust variance, as this explanation will provide key information for the reader. For a useful reference on this methodological decision.

See:

Thompson ML, Myers JE, Kriebel D. Prevalence odds ratio or prevalence ratio in the analysis of cross sectional data: what is to be done? Occup Environ Med. 1998 Apr;55(4):272-7. doi: 10.1136/oem.55.4.272.

McNutt LA, Wu C, Xue X, Hafner JP. Estimación del riesgo relativo en estudios de cohortes y ensayos clínicos de resultados comunes. Am J Epidemiol. 2003 May 15;157(10):940-3. doi: 10.1093/aje/kwg074.

• We understand the concern. In our study, we selected binary logistic regression to evaluate predictors of QFT−TST− at baseline and at three months, rather than employing a Poisson regression with robust variance. This methodological choice was guided by the nature of the outcome variables, the research objectives, and the properties of logistic regression, as outlined below:

o Nature of the Outcome Variables:

The primary outcomes in this study were binary variables (QFT+TST+ and QFT−TST-), for which logistic regression is the most commonly applied and well-established modelling approach.

o Prevalence of the Outcome Variables:

While it is acknowledged that odds ratios may not approximate prevalence ratios when the outcome prevalence exceeds 10% (Thompson et al., 1998), the primary focus of this analysis was not to estimate prevalence or prevalence ratios, but rather to identify significant predictors of QFT−TST− and their relative contributions to the outcome. Logistic regression remains appropriate in this context, as it effectively assesses associations between predictors and outcomes while controlling for confounders.

o Assessment of Model Fit and Clustering Effects:

Prior to employing logistic regression, we evaluated the need to account for potential clustering due to the sampling distribution. This was achieved by fitting an intercept-only mixed-effects model and comparing it to a standard binary logistic regression model. With a chi-square test p-value of 0.4, indicating no significant improvement in fit when accounting for clustering, we proceeded with the logistic regression model for subsequent analyses. We have also added text in the methodology section that “coupled with the primary objective of identifying significant predictors associated with remaining QFT-TST- at baseline and at three months follow-up despite TB exposure (Lines 196-198)”

o Comparison with Poisson Regression with Robust Variance:

We acknowledge that Poisson regression with robust variance is an alternative approach for modelling binary outcomes, particularly when estimating prevalence ratios for common outcomes (McNutt et al., 2003). However, the assumptions and focus of Poisson regression are more suited to studies prioritizing prevalence ratio estimation, which was not the primary goal of this study. Instead, logistic regression was selected for its flexibility and interpretability in identifying predictors and quantifying their associations with the outcomes of interest.

o Also, most of the manuscripts with similar study designs deployed logistic regression especially when there was an absence of clustering in their sample population (Ma et al [7], Mave et al [8] and Verrall et al [9]).

3. I suggest including in the methodology section a clear specification of ‘VARIABLES’: independent variable, dependent variable, and adjustment or confounding variables. In addition, it would be advisable to add a footnote to the corresponding table, indicating that certain variables are not interpretable and including a brief justification. This will provide greater clarity to the reader and reduce the risk of misinterpretation regarding the role of the adjustment variables.

You may find it useful to consult the following reference:

• Thank you for your valuable comment; by incorporating all variables, we aimed to minimize the risk of omitted-variable bias and allow for a comprehensive assessment of possible associations, which also in accordance with previous literature (Ma et al [7], Mave et al [8] and Verrall et al [9]). However, we acknowledge that including multiple variables in the model does not imply direct causality. To address potential concerns related to misinterpretation of adjusted estimates, we have updated Table 3, Table 4 and the supplementary Table 1 with a clear footnote explaining that the associations reported should not be interpreted as causal effects but rather as potential factors for further investigation.

Akinkugbe AA, Simon AM, Brody ER. A scoping review of Table 2 fallacy in the oral health literature. Community Dent Oral Epidemiol. 2021 Apr;49(2):103-109. doi: 10.1111/cdoe.12617.

---

## [Decision Letter · Decision Letter 1]

3 Mar 2025

PONE-D-24-20846R1Epidemiological factors associated with immunological resistance in household contacts exposed to active tuberculosis in South Africa: an analysis using multivariable regressionPLOS ONE

Dear Dr. Maenetje,

Thank you for submitting your manuscript to PLOS ONE. After careful consideration, we feel that it has merit but does not fully meet PLOS ONE’s publication criteria as it currently stands. Therefore, we invite you to submit a revised version of the manuscript that addresses the points raised during the review process.

We look forward to receiving your revised manuscript.

Kind regards,

Xiangwei Li

Academic Editor

PLOS ONE

Reviewers' comments:

Reviewer's Responses to Questions

**Comments to the Author**

1. If the authors have adequately addressed your comments raised in a previous round of review and you feel that this manuscript is now acceptable for publication, you may indicate that here to bypass the “Comments to the Author” section, enter your conflict of interest statement in the “Confidential to Editor” section, and submit your "Accept" recommendation.

Reviewer #1: All comments have been addressed

Reviewer #3: All comments have been addressed

2. Is the manuscript technically sound, and do the data support the conclusions?

Reviewer #1: Yes

Reviewer #3: Yes

3. Has the statistical analysis been performed appropriately and rigorously? 

Reviewer #1: Yes

Reviewer #3: Yes

4. Have the authors made all data underlying the findings in their manuscript fully available?

Reviewer #1: Yes

Reviewer #3: Yes

5. Is the manuscript presented in an intelligible fashion and written in standard English?

Reviewer #1: Yes

Reviewer #3: Yes

6. Review Comments to the Author

**Reviewer #1:**  I thank the authors for considering my comments and correcting the manuscript accordingly. I have no additional comments.

**Reviewer #3:**  Given that the author justifies the use of binary logistic regression on the basis of previous studies with similar methodological approaches, but acknowledges that odds ratios may not be an adequate proxy for prevalence ratios when the prevalence of the outcome exceeds 10% (Thompson et al., 1998), it is necessary to readjust the title of the study to SPECIFY precisely the type of regression used.

This change would ensure greater methodological rigour and avoid possible misinterpretation by statistically trained readers who might assume a different analytical approach.

Suggestion:

De:

‘Epidemiological factors associated with immune resistance in household contacts exposed to active tuberculosis in South Africa: a multivariate regression analysis’.

A:

‘Epidemiological factors associated with immune resistance in household contacts exposed to active tuberculosis in South Africa: a logistic regression analysis’.

7. PLOS authors have the option to publish the peer review history of their article (what does this mean? ). If published, this will include your full peer review and any attached files.

**Do you want your identity to be public for this peer review?** For information about this choice, including consent withdrawal, please see our Privacy Policy .

Reviewer #1: No

Reviewer #3: **Yes: ** Huamani-Echaccaya Jose Luis

---

## [Author Response · Author response to Decision Letter 2]

6 Mar 2025

Reviewer #1:

I thank the authors for considering my comments and correcting the manuscript accordingly. I have no additional comments.

• We thank the reviewer for accepting the amendment

Reviewer #3:

Given that the author justifies the use of binary logistic regression on the basis of previous studies with similar methodological approaches, but acknowledges that odds ratios may not be an adequate proxy for prevalence ratios when the prevalence of the outcome exceeds 10% (Thompson et al., 1998), it is necessary to readjust the title of the study to SPECIFY precisely the type of regression used.

This change would ensure greater methodological rigour and avoid possible misinterpretation by statistically trained readers who might assume a different analytical approach.

Suggestion:

De:

‘Epidemiological factors associated with immune resistance in household contacts exposed to active tuberculosis in South Africa: a multivariate regression analysis’.

A:

‘Epidemiological factors associated with immune resistance in household contacts exposed to active tuberculosis in South Africa: a logistic regression analysis’.

• We thank the reviewer for this comment and have updated the manuscript title as requested.

---

## [Decision Letter · Decision Letter 2]

18 Jul 2025

Epidemiological factors associated with immunological resistance in household contacts exposed to active tuberculosis in South Africa: a logistic regression analysis

PONE-D-24-20846R2

Dear Dr. Maenetje,

We’re pleased to inform you that your manuscript has been judged scientifically suitable for publication and will be formally accepted for publication once it meets all outstanding technical requirements.

Kind regards,

Frederick Quinn

Academic Editor

PLOS ONE

Additional Editor Comments (optional):

Reviewers' comments:

Reviewer's Responses to Questions

**Comments to the Author**

1. If the authors have adequately addressed your comments raised in a previous round of review and you feel that this manuscript is now acceptable for publication, you may indicate that here to bypass the “Comments to the Author” section, enter your conflict of interest statement in the “Confidential to Editor” section, and submit your "Accept" recommendation.

Reviewer #1: All comments have been addressed

Reviewer #3: All comments have been addressed

2. Is the manuscript technically sound, and do the data support the conclusions?

Reviewer #1: (No Response)

Reviewer #3: Yes

3. Has the statistical analysis been performed appropriately and rigorously? 

Reviewer #1: (No Response)

Reviewer #3: Yes

4. Have the authors made all data underlying the findings in their manuscript fully available?

Reviewer #1: (No Response)

Reviewer #3: Yes

5. Is the manuscript presented in an intelligible fashion and written in standard English?

Reviewer #1: (No Response)

Reviewer #3: Yes

6. Review Comments to the Author

Reviewer #1: (No Response)

Reviewer #3: It was verified that the authors have taken into account the observations and suggestions made in the previous review, making the necessary adjustments to both the content and structure of the manuscript. The modifications incorporated contribute to greater clarity, coherence, and scientific rigour in the work, which demonstrates the authors' willingness to improve the quality of the article.

7. PLOS authors have the option to publish the peer review history of their article (what does this mean? ). If published, this will include your full peer review and any attached files.

**Do you want your identity to be public for this peer review?** For information about this choice, including consent withdrawal, please see our Privacy Policy .

Reviewer #1: No

Reviewer #3: **Yes: ** Huamani-Echaccaya Jose Luis

---

## [Editor Report · Acceptance letter]

PONE-D-24-20846R2

PLOS ONE

Dear Dr. Maenetje,

I'm pleased to inform you that your manuscript has been deemed suitable for publication in PLOS ONE. Congratulations! Your manuscript is now being handed over to our production team.

Kind regards,

on behalf of

Dr. Frederick Quinn

Academic Editor

PLOS ONE